# Recommendations for evaluating photoplethysmography-based algorithms for blood pressure assessment

Mohamed Elgendi [1] ✉, Fridolin Haugg[1], Richard Ribon Fletcher[2], John Allen [3], Hangsik Shin [4], Aymen Alian[5] & Carlo Menon [1]

Photoplethysmography (PPG) is a non-invasive optical technique that measures changes in blood volume in the microvascular tissue bed of the body. While it shows potential as a clinical tool for blood pressure (BP) assessment and hypertension management, several sources of error can affect its performance. One such source is the PPG-based algorithm, which can lead to measurement bias and inaccuracy. Here, we review seven widely used measures to assess PPG-based algorithm performance and recommend implementing standardized error evaluation steps in their development. This standardization can reduce bias and improve the reliability and accuracy of PPG-based BP estimation, leading to better health outcomes for patients managing hypertension.

## Photoplethysmography and blood pressure

Photoplethysmography (PPG) technology[1], while not yet the gold standard for blood pressure (BP) monitoring, offers innovative solutions to some of the limitations faced by traditional methods in managing hypertension. One of the most promising aspects of PPG is its potential to enable beat-to-beat BP monitoring, a capability not feasible with current gold-standard techniques[2]. Traditionally, BP measurement relies on sphygmomanometers, which, though accurate, are limited to discrete measurements and can cause discomfort during use[3]. In contrast, PPG offers a less invasive approach, allowing for continuous monitoring of BP[4]. This makes PPG an appealing alternative for both patients and healthcare providers, as it opens up the possibility for more detailed and responsive hypertension management.

Hypertension, a condition characterized by persistently high BP, poses significant health risks if not managed effectively. Individuals with normal BP are considered normotensive, while those with lower-than-normal readings are classified as hypotensive. Managing these conditions requires accurate and timely monitoring, a challenge that PPG technology aims to address with its ability to offer continuous, real-time data through both contact[5] and contactless[6] sensors.

The PPG waveform, a visual representation of blood volume changes within the microvascular tissue bed, provides essential insights into cardiovascular health. PPG devices, capable of both contact and contactless measurements, can estimate BP and other cardiovascular parameters.

Contact PPG devices include not only wearable technologies like smartwatches and fitness trackers, which require direct skin contact using a light source and photodetector to collect data, but also textile-based sensors. These textile sensors are integrated into everyday clothing, offering a unique approach to BP monitoring by maintaining direct contact in a more comfortable and flexible manner[7]. This innovation has recently gained attention for its non-invasiveness and convenience, allowing for continuous BP monitoring during daily activities. As the field of PPG technology evolves, the development of contactless methods opens new possibilities for even more unobtrusive health monitoring.

Contactless PPG technology is still in the early stages of development but has the potential to provide a more convenient and non-invasive way to measure BP. Contactless PPG devices employs methods to measure changes in blood volume without direct contact with the skin. Different types of contactless PPG devices exist, such as camera-based and radar-based systems. Camera-based systems[8] use a camera to capture the image of the face or body limbs finger and then analyze the pulsatile changes in skin color to estimate blood flow and BP. Doppler radar-based systems[9] are very sensitive motion sensors that can detect pulsatile activity in order to estimate changes in blood volume and estimate BP.

The development of PPG-based BP detection models represents a promising advancement in non-invasive healthcare monitoring. This multi-step process starts with the collection of PPG data, which records blood volume changes using optical methods[1,5]. Such data is pivotal for identifying

[1]Biomedical and Mobile Health Technology Lab, Department of Health Sciences and Technology, ETH Zurich, Zurich 8008, Switzerland. [2]Mechanical Engineering, Massachusetts Institute of Technology, Cambridge, MA 02139, USA. [3]Research Centre for Intelligent Healthcare, Coventry University, CV1 5FB Coventry, UK. [4]Department of Convergence Medicine, Asan Medical Center, University of Ulsan College of Medicine, Seoul 05505, Republic of Korea. [5]Yale School of Medicine, Yale University, New Haven, CT 06510, USA. ✉e-mail: moe.elgendi@hest.ethz.ch

patterns (also referred to as features) that correlate with blood pressure fluctuations[10,11]. Recognizing these patterns is crucial for the subsequent development of mathematical models, including statistical, machine learning, or deep learning approaches. These models are adept at performing a range of critical functions: classifying individuals as normotensive, hypertensive, or hypotensive;[10,11] and accurately estimating BP values[12]. This capability to categorize BP readings not only streamlines the identification of health risks but also positions PPG as an invaluable tool in the landscape of blood pressure monitoring.

However, the reliability of PPG-based BP models must be rigorously evaluated in diverse patient cohorts to ensure their generalizability, a prerequisite for their approval by regulatory bodies and eventual clinical application. Recent literature reviews have highlighted potential inaccuracies in these models' predictive abilities, often resulting in measurement biases that either overestimate or underestimate BP values[13,14]. These discrepancies may stem from varying evaluation standards, performance metrics, and the representativeness of datasets used in PPG analysis, which sometimes exhibit biases in demographic characteristics like gender or age[15].

Below, we discuss what requirements need to be met to reduce model (or algorithm) bias and explore the array of performance metrics employed in evaluating PPG-based algorithms for BP measurement, as documented in existing literature. Our analysis leads us to recommend seven distinct approaches for assessing BP measurement performance, encompassing six quantitative metrics alongside a visual tool. These methodologies are designed to offer a holistic evaluation framework, ensuring a comprehensive performance assessment. By employing these approaches, we aim to enhance the comparability of PPG-based models with current clinical standards, thereby bolstering their reliability and reducing inherent biases. This multifaceted evaluation strategy not only underscores the importance of precision and accuracy in BP monitoring but also highlights the potential of PPG technology in revolutionizing hypertension management.

## The importance of diverse PPG datasets

The dataset used should be assessed to ensure that it has a sufficient size, and that it is balanced and unbiased. The most important attributes of datasets for PPG-based BP models are: demographic data (e.g., gender, age, weight, and height), BP distribution (i.e., hypotensive, normotensive, or hypertensive), skin color, and number of people. The correlation between demographic data and BP can potentially be very high, and a poor model may achieve good results, but only when using small, homogeneous, and non-representative datasets. Several PPG-based models use only datasets with normotensive people, which simplifies the determination of BP and usually leads to a high score in the applied performance measure[16,17]. Since PPG models use reflected light from the skin, the distribution of skin tones must also be considered. Several camera-based PPG models are limited to be applied mainly to Asian people; therefore, the results cannot be generalized to the wider population[18,19].

For machine learning models, the size of the dataset used is of particular importance, as they are data driven. In addition, only a small percentage (approximately 10−20%) of the whole dataset can be used for testing since the majority is used for training. Note that highlighting models as 'data-driven' underlines the critical importance of utilizing large, balanced, and unbiased datasets. The reliability and precision of machine learning models in estimating blood pressure are fundamentally contingent upon their training with a broad and representative array of real-world data. Therefore, it is essential to ensure that the datasets employed in the development of PPG-based BP models adhere to these standards. This adherence is pivotal to enhancing their effectiveness and ensuring their practical utility in clinical environments.

Incorporating special populations, such as pregnant women, into PPG-based BP model development is also crucial. Pregnancy-induced hypertension, a common condition, poses significant health risks to both mother and fetus. The unique BP measurement challenges in this demographic are not adequately addressed by current standards for automated BP devices. Including pregnant women in research datasets is imperative for creating

BP assessment models that are both accurate and reliable for their specific needs.

## PPG-based BP calibration

Calibration is a pivotal step in the development of PPG-based BP devices, ensuring the accuracy and reliability of measurements provided by these non-invasive technologies[5,20]. Calibration forges a definitive relationship between the PPG signal captured by the device and reference BP measurements, which are traditionally obtained through either direct invasive methods, such as arterial BP lines (a cannula inserted into a peripheral artery), or non-invasively, via cuff-based sphygmomanometry. The development of the calibration model, or algorithm, leverages reference BP measurements in conjunction with the PPG signals captured by the device. This process employs diverse computational techniques, including regression models, machine learning algorithms, and signal processing methods. These techniques are utilized to mathematically correlate the extracted features of the PPG waveform with BP readings. This significant research effort aims to create algorithms that effectively translate PPG data into precise BP values or classify BP status using specific PPG features[11,21].

Moreover, the accuracy of BP estimates and the calibration process for PPG-based devices are significantly influenced by three key factors. Firstly, the measurement site for PPG signal measurement (e.g., finger, wrist, or earlobe) significantly impacts the PPG waveform's morphology, primarily due to variations in pulse transit time (PTT). The PTT is the time it takes for the arterial pulse wave to travel from the heart to the measurement site, which is integral to the waveform analysis. PTT is utilized as a crucial input in calibration models, given its strong correlation with BP levels, highlighting the importance of accurately measuring and interpreting PTT in BP estimation[5,22]. Secondly, vessel compliance, or the elasticity of blood vessels, influences PTT measurements, thus affecting BP estimation accuracy[23]. Thirdly, hydrostatic pressure and gravitational effects play a role. The level of the measurement site relative to the heart introduces variations in BP readings due to hydrostatic pressure differences and gravitational effects on blood flow. This factor necessitates adjustments in calibration models to account for the elevation or depression of the measurement site in relation to the heart's position[24].

To counteract these effects, calibration models incorporate additional features or correction factors. These ensure that BP measurements are normalized for the measurement site's distance from the heart and its relative level, enhancing accuracy. The calibration process is crucial for transforming theoretical models and algorithms into practical, reliable PPG-based BP devices. This comprehensive approach, from selecting the measurement site to calibrating devices against reference measurements, ensures the devices are fit for everyday BP monitoring.

## Calibration strategies for continuous monitoring

In the realm of PPG-based BP monitoring, calibration strategies play a pivotal role in ensuring the accuracy and reliability of continuous monitoring systems. These strategies can broadly be categorized into subject-specific and non-subject-specific approaches[25]. Subject-specific calibration tailors the calibration process to individual users by aligning paired recordings of BP and PPG data. This customization allows algorithms to generate precise BP estimations for that individual, enhancing subsequent continuous monitoring's accuracy. While this approach achieves high accuracy, its practicality and cost-effectiveness are limited for widespread application.

Non-subject-specific calibration uses data from a broad population, occasionally incorporating PTT measurements, to develop a generalized model. Though this method might not match the accuracy of subject-specific models, it balances the need for scalability with reasonable accuracy. To address these limitations, hybrid calibration strategies have emerged, combining individual calibration data points with demographic information and BP-PPG pairs from diverse populations. This approach enhances accuracy and generalizability without losing the practicality of non-subject-specific methods[25]. Hybrid calibration requires minimal individual data,

relying mainly on demographics and a comprehensive training set for parameter estimation. Despite the influence of factors like age, sex, and BMI on PPG waveforms, population calibration-using only demographic data and no individual calibration recordings-typically lacks sufficient accuracy[26]. Consequently, hybrid strategies are favored in recent PPG-based BP estimation studies for their ability to offer both precision and broad application[26].

Emerging from this challenge are hybrid calibration strategies, which innovatively combine the strengths of both subject-specific and non-subject-specific approaches. These strategies utilize a composite dataset that includes both individualized data and general population data, potentially supplemented by PTT measurements. The hybrid model aims to leverage the detailed accuracy obtained from subject-specific data while enhancing the model's robustness and applicability through insights drawn from the broader population data. This amalgamation results in a calibration model that does not merely offer a compromise but seeks to optimize accuracy and scalability. Such an approach enhances the generalizability of the calibration model, ensuring it remains practical for widespread implementation without significantly sacrificing individualized precision[25].

Continuous PPG-based BP tracking, which records BP trends over time, relies heavily on these calibration models to provide meaningful insights. Devices and applications implementing this technology, from wearable fitness devices to smartphone apps, capture PPG signals to monitor BP changes relative to baseline readings. Studies, such as the one by Radha et al.[27], demonstrate the potential of PPG technology in tracking nocturnal systolic BP, highlighting the importance of frequent measurements. However, the precision of these measurements at any given moment can vary, underscoring the necessity for well-calibrated devices in clinical diagnosis and treatment planning.

Transitioning from BP tracking to BP measurement illustrates the critical importance of calibration in both contexts. BP measurement with PPG aims to obtain precise and accurate single-point BP values, requiring validated and calibrated devices like mercury sphygmomanometers or arterial BP monitors for accurate systolic and diastolic BP readings. The calibration process, whether subject-specific as explored by Tang et al.[28] or non-subject-specific as shown by Ibtehaz et al.[29], is fundamental in mitigating errors and biases. These approaches, despite their potential, highlight the ongoing challenges in real-time application and the continuous need for improvement in calibration techniques to ensure PPG-based BP devices' effectiveness in both monitoring and measuring BP.

Incorporating these calibration strategies and understanding their implications on the accuracy and reliability of PPG-based BP devices is essential. It addresses the critical feedback regarding the linkage between device applications and calibration, and how they collectively contribute to minimizing error and bias in BP measurement. By situating this discussion within the broader context of state-of-the-art PPG applications for BP measurement, it becomes clear that calibration is not just a technical requirement but a cornerstone in the development and application of PPG technology for health monitoring.

## Performance measures and their calculations

After discussing strategies to minimize bias and enhance accuracy through careful dataset selection and rigorous calibration techniques, it's essential to evaluate how these efforts translate into the performance of PPG-based BP measurement devices. Performance metrics serve as crucial tools in this assessment, providing quantifiable measures to gauge the effectiveness of bias mitigation and calibration accuracy. The following performance measures (Table 1) are applied to assess both systolic and diastolic BP derived from PPG signals, offering insights into the precision and reliability of the measurements.

Among the key performance metrics for evaluating PPG-based BP measurements are the Mean Error (ME) and the Standard Deviation of Error (SDE). ME represents the average deviation of estimated BP values from true values, serving as a direct measure of accuracy. However, relying solely on ME as a performance measure is insufficient. This is because ME calculates the mean of all errors, allowing positive and negative deviations to

**Table 1 | Seven evaluation measures (six quantitative metrics and one visualization plot) used to examine the performance of PPG-based blood pressure algorithms**

| Performance measure | Equation |
|---|---|
| Mean absolute error (MAE) | $\frac{1}{n}\sum_{i=1}^{n}|x_i - y_i|$ |
| Mean error (ME) | $\frac{1}{n}\sum_{i=1}^{n} x_i - y_i$ |
| Standard deviation of error (SDE) | $\sqrt{\frac{1}{n-1}\sum_{i=1}^{n}\left(x_i - y_i - \frac{1}{n}\sum_{i=1}^{n} x_i - y_i\right)^2}$ |
| Root mean squared error (RMSE) | $\sqrt{\frac{1}{n}\sum_{i=1}^{n}(x_i - y_i)^2}$ |
| Pearson's correlation coefficient (R) | $\frac{\sum_{i=1}^{n} x_i y_i - \sum_{i=1}^{n} x_i \sum_{i=1}^{n} y_i}{\sqrt{n\sum_{i=1}^{n} x_i^2 - \left(\sum_{i=1}^{n} y_i\right)^2}\sqrt{n\sum_{i=1}^{n} y_i^2 - \left(\sum_{i=1}^{n} y_i\right)^2}}$ |
| Absolute error of sample $i$ | $|x_i - y_i|$ |
| Bland–Altman plot | $\left(\frac{x_i + y_i}{2}, x_i - y_i\right)$ |

$n$ number of samples, $x_i$ estimated value, $y_i$ reference value.

offset one another, potentially resulting in a misleadingly small ME value[30]. This scenario can occur even if the model frequently makes large errors in both underestimation and overestimation, thereby concealing the true variability and inaccuracy of the BP estimates.

Providing the error a quantitative metric as ME ± SDE is useful for describing the errors' distribution and assessing the model's performance. In the case of machine learning models, an ME value near zero can easily be achieved without identifying the relationship between PPG and BP by predicting the average value of the dataset. SDE is also included to describe the variability and spread in the BP values, which measures how widely the estimated BP values are dispersed providing the composite error metric as ME ± SDE is thus useful in describing the distribution of errors for assessing the model's performance. Together, the ME and SDE can provide an indication of the accuracy and precision of the measurement device. A small ME and SDE suggest that the measurements are not only close to the true BP value but also demonstrate high repeatability across multiple measurements. Conversely, a large ME and SDE signify that the BP measurements deviate more from the true value and exhibit greater variability, thereby indicating lower repeatability. This concept underscores the importance of achieving minimal variability in measurements to ensure their reliability and consistency over time.

The Mean Absolute Error (MAE) is a commonly utilized performance metric, especially in the evaluation of ML models and mathematical models for PPG-based BP measurement. MAE quantifies the average absolute deviation of estimated BP values from their true values, offering a straightforward measure of a model's accuracy. It is determined by calculating the mean of the absolute differences between each estimated measurement and its corresponding true value. A key advantage of MAE is its indifference to the direction of errors, meaning it does not differentiate between overestimations or underestimations. This characteristic makes MAE particularly valuable for assessing the overall magnitude of error in BP measurement models, providing a clear indication of how closely the model's predictions align with actual BP readings.

The root mean square error (RMSE) represents the square root of the differences between the estimated BP and the reference BP or the quadratic mean of these differences. In other words, ME measures the average deviation of the measurements from the true value, SDE measures the variability or spread of the error in the measurements, MAE measures the average absolute deviation of the measurements from the true value, and RMSE measures the average magnitude of the error.

According to O'Brien et al.[31], for clinical acceptance, it is essential to report the number of absolute errors within specific thresholds (5, 10, and 15 mmHg) alongside utilizing Bland–Altman plots for a comprehensive analysis. For instance, in a scenario where five reference measurements are paired with five BP estimates, resulting in absolute errors of 7 mmHg, 3 mmHg, 6 mmHg, 2 mmHg, and 11 mmHg, one can determine the proportion of measurements falling within clinically acceptable error margins.

This analysis highlights the accuracy of BP estimates in real-world conditions, demonstrating two measurements within a 5 mmHg error, four within 10 mmHg, and all five within 15 mmHg. The Bland–Altman plot, a tool for assessing the agreement between two measurement techniques, extends this analysis by visualizing the difference between paired BP readings against their average. Unlike ME and MAE, which quantify error magnitudes, Bland–Altman plots[32] provide insights into the consistency and bias between the compared methods, using two SDE lines to delineate limits of agreement. This method's relevance extends beyond PPG devices; it is also recommended for evaluating non-invasive inflatable cuff-based sphygmomanometers[33]. This comparison is crucial because it positions the performance of PPG-based devices against the backdrop of broader BP measurement technologies. It emphasizes the necessity of employing standardized evaluation methods, such as the Bland–Altman plot, to guarantee the reliability and clinical relevance of all BP monitoring tools, including those utilizing cutting-edge PPG technology.

Pearson's correlation coefficient (R) measures the relationship between measured BPs from a reference device (e.g., arterial BP) and the BP estimated from the PPG-based BP values in continuous monitoring. It quantifies the strength and direction of the linear relationship, indicating the level of agreement between PPG-based BP estimates and the reference measurements. Calculating the correlation coefficient allows for assessing the association between the two BP measurements in the time series data. While R measures the linear relationship between two variables, it may not capture all aspects of agreement, especially if the relationship is nonlinear or there are systematic differences between the methods[34]. In such cases, alternative metrics like Bland–Altman analysis may provide more comprehensive insights into the agreement between clinical measurement methods.

## The universal standard for the validation of BP measuring devices

A variety of international protocols exist for the evaluation and validation of automated sphygmomanometers. The 2018 Universal Standard for the Validation of Automated BP Measurement Devices, which was developed by the American Association for the Advancement of Medical Instrumentation (AAMI), the European Society of Hypertension (ESH), and the International Organization for Standardization (ISO), is one of the most recent summary reports to be accepted for validation[35]. This summary report outlines nine essential components of the validation procedure that were agreed upon by all the representatives from the AAMI, ESH, and ISO as common ground for designing a single universal protocol for the validation of BP monitors. The objective of this report is not to offer a thorough description of all parts of the validation procedure; nevertheless, when the organizations involved develop the procedural details of the universal protocol, this information will become available. This protocol is intended to be used for the design of cuff-based applications, but it has also been employed in several studies to validate cuffless PPG-based BP models, and several aspects that can be applied to these models.

An AAMI/ISO/ESH general population validation study requires 85 participants over 12 years old, who may or may not have been untreated or treated for hypertension. The source for which both hypertensive and normotensive subjects were recruited should be reported. The study should comprise $n = 85$ adults that include ≥30% males and ≥30% females, with ≥5% of the subjects within the reference systolic BP readings (≤100 mmHg), ≥5% with ≥160 mmHg, and ≥20% with ≥140 mmHg. Furthermore, ≥5% of the subjects should be within the reference diastolic BP readings (≤60 mmHg), ≥5% with ≥100 mmHg and ≥20% with ≥85 mmHg.

Mercury sphygmomanometers or accurate non-mercury devices should be used to measure the reference BP. The reference values were obtained by averaging the readings of two trained observers making simultaneous non-invasive BP measurements on each subject using the auscultatory technique and a double stethoscope[33]. The accuracy of non-mercury devices must be evaluated at the beginning of each study. The ME should be within or equal to ±5 mmHg, and the SDE should not be greater than 8 mmHg for systolic BP and diastolic BP. The number of absolute BP differences within 5, 10, and 15 mmHg and standardized Bland–Altman scatterplots should be presented for SBP and DBP[31]. This is only a brief description of the essential components of the AAMI/ISO/ESH summary report that could be used for PPG-based approaches. This summary report announces the development of separate universally applicable protocols for cuffless BP measurements. The most current standard for non-invasive sphygmomanometers is the third edition of ISO 81060-2:201811[33]. This standard is used as a basis and may then be adapted in a specific country. However, differences exist in the name of the standard (e.g., adding a country-specific acronym, such as AAMI or EN before ISO 81060-2:201811) and the introduction, mainly due to the need to explain connections with other standards. The actual standard text is the same, so ISO 81060-2:2018[33] is sufficient for technical purposes. Nevertheless, the standard is intended for sphygmomanometers and not for PPG-based applications that measure BP over a usually longer time interval. The requirements of this standard are not usually applied to PPG-based BP monitoring studies, as it requires a considerable amount of effort to be met. For example, the protocol requires at least three valid paired BP values for each subject. Nevertheless, individual aspects can serve as a guideline, as described above.

## Current PPG-based BP measurement research status

To evaluate which performance measures are commonly used, we analyzed the reviews by Chan et al.[14] and Hosanee et al.[13]. The focus of Chan et al.'s[14] review paper is on understanding the characteristics of PPG associated with BP and the evolution of this technology from 2010 to 2019 in terms of validation, sample size, diversity of individuals, and datasets employed. A total of 25 publications were evaluated, and the study exclusively reports single-site technologies, meaning that the PPG signal is only evaluated from a single-body location, and it is measured pointwise. The review by Hosanee et al.[13] focused on a total of 13 publications that used multi-site PPG.

One key aspect measured by multi-site PPG is arterial pulse wave velocity (PWV). Arterial PWV is defined as the speed at which the arterial pulse wave travels between two points on the body. It is a crucial parameter for assessing arterial stiffness and can be indicative of cardiovascular health. By measuring the time it takes for the pulse wave to travel from one site to another, multi-site PPG can continuously estimate BP, leveraging the relationship between PWV, PTT, and BP.

### Sample size

As discussed above, the number of subjects is one of the most important evaluation criteria for the dataset. Figure 1 depicts what percentage of the evaluated studies used a certain number of subjects for single-site and multi-site PPG-based models, and how many studies met the requirement for the minimum number of subjects from the AAMI/ISO/ESH summary report[35] for single-site and multi-site PPG-based models. Only six out of the 25 single-site studies used more than 85 subjects. However, in four of the six publications[17,36–38] that applied machine learning models, only a small portion of the dataset (approximately 10−20%) was used for testing the machine learning model. According to Chan et al.[14], only 8% of the reviewed studies conducted their experiments on a participant pool that included hypertensive patients, and only 4% included hypotensive subjects. None of the 13 studies on multi-side models used over 85 subjects. According to Hosanee et al.[13], most studies used either healthy subjects with no hypertension, nor those with co-morbidities, nor pregnant women.

Alternatively, they used health statuses that were undisclosed. In both reviews, the studies assessed often used too few subjects to meet the AAMI/ISO/ESH summary report requirements from 2018[35]. In addition, hypertensive or hypotensive subjects were often not included. This is unfortunate because early detection of an abnormal BP condition is essential for timely treatment. Early detection of abnormal BP is crucial in preventing or managing various health conditions such as hypertension, heart disease, stroke, and kidney disease. A model that can only determine normal BP has very limited applicability.

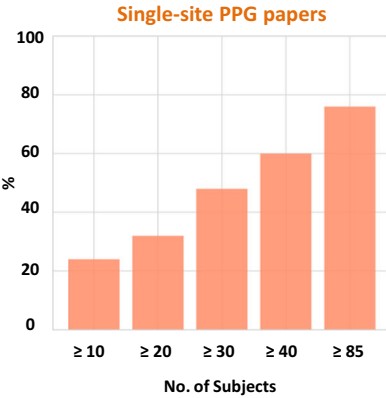

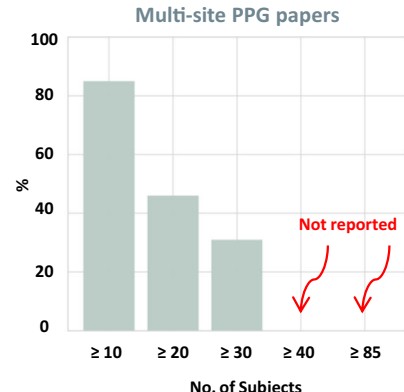

**Fig. 1 | Analysis of subject numbers in PPG studies.** This chart displays the percentage of studies reporting different ranges of subjects used, categorized into multi-site and single-site PPG studies. The *x*-axis represents the number of subjects reported, while the *y*-axis indicates the percentage of studies. Ranges shown include ≥10, ≥20, ≥30, ≥40, and ≥85 subjects, with a separate category for studies that did not report the number of subjects. The left figure presents the results for single-site PPG studies (totaling 25 models), while the right figure shows results for multi-site PPG studies (totaling 13 models). Categories not reported in the studies are indicated in red. Notably, no multi-site PPG study complied with the 85-subject minimum recommended by the AAMI/ISO/ESH summary report[35]. The data used to create this figure can be found in Supplementary Data 1.

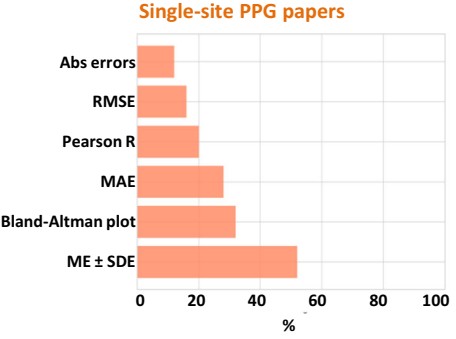

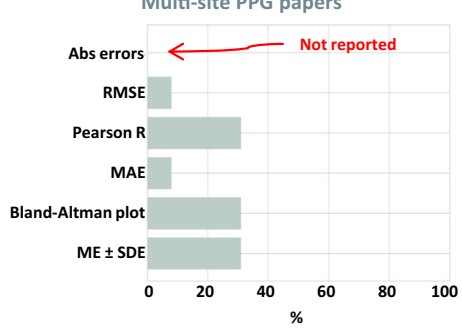

**Fig. 2 | Analysis of performance measures in PPG studies.** This figure compares the performance measures used in single-site and multi-site PPG studies. The measures analyzed include ME ± SDE (Mean Error ± Standard Deviation of Error), Mean Absolute Error (MAE), Bland–Altman plot, Pearson correlation coefficient (Pearson R), Root Mean Square Error (RMSE), and Absolute Errors (Abs errors). The left figure presents the results for single-site PPG studies (a total of 25 models), while the right figure shows the results for multi-site PPG studies (a total of 13 models). The number of absolute errors within 5, 10, and 15 mmHg, as required by the AAMI/ISO/ESH summary report[35], was the least-reported measure, appearing in 13% of all single-site studies and never reported in multi-site studies (indicated in red). The Bland–Altman plot was used with approximately the same frequency in single-site and multi-site models, with 68% of single-site studies and 69% of multi-site studies not using this measure. The data used to create this figure can be found in Supplementary Data 1.

## Performance measures

ME ± SDE was the most frequently used measure for single-side models and was applied by 52% of the studies reviewed. This makes sense, given that the ME ± SDE is also mentioned in the AAMI/ISO/ESH summary report[35]. However, this also means that only about half of the studies can be directly compared with the same performance measures. The AAMI/ISO/ESH summary report[35] also requires a Bland–Altman plot and numbers of absolute errors of over 5, 10, and 15 mmHg by utilizing an accepted clinical protocol[31], this requirement was met in 32% of the studies using single-site PPG models and in 11% of the studies employing multi-site PPG models. Furthermore, our analysis found that the MAE and R were consistently utilized across the studies, appearing in 28% and 24% of them, respectively. Additionally, the RMSE was reported in 16% of the studies reviewed by both Chan et al.[14] and Hosanee et al.[13].

Figure 2 depicts the percentages of the evaluated studies that used a particular measure for single-site and multi-site PPG-based models. As we can see, there are inconsistencies in evaluating PPG-based BP algorithms. The optimal case would be a full circle for all evaluation measures, equivalent to 100%. In all PPG-based studies, the Bland–Altman plot with Mean Error (ME) was employed more often compared to MAE. Due to the lack of a consistent performance measure across all studies, the variety of metrics employed makes comparisons between them challenging.

In our analysis, we observed notable gaps in the current research landscape for PPG-based BP measurement, particularly the scarcity of comprehensive PPG databases and a lack of thorough failure analyses. These gaps highlight the need for more robust data collection and evaluation methodologies to enhance the reliability and applicability of PPG technology in diverse populations.

## Recommendations and call to action

After addressing the aforementioned gaps and considerations, we recommend several actions to advance the development of PPG-based BP models. Firstly, it is essential to report results using the seven performance measures: MAE, ME, SDE, RMSE, R, absolute error of sample, and Bland–Altman plot. Additionally, reporting results using ME ± SDE as a performance measure is crucial, as ME ± SDE is also used in the collaborative statements issued by AAMI, ESH, and ISO[33].

Furthermore, creating a comprehensive dataset collected from representative ages, genders, and ethnicities, each with a broad spectrum BP distribution (hypotensive, normotensive, and hypertensive), following the guidelines of the consensus document from AAMI, ESH, and ISO for the general population, is highly recommended[35]. It is also important to include pregnant women in the dataset used for the PPG-based BP model. After reviewing all current standards for the validation of automated BP

measurement devices, none of them addressed the specific challenges and considerations for measuring BP in pregnant women.

Our recommendations, in conjunction with the 2023 ESH statement[39], offer a comprehensive framework for validating cuffless BP measuring devices. The ESH statement introduces six validation tests to assess crucial aspects such as absolute BP accuracy, robustness against hydrostatic pressure effects, accuracy during treatment, awake/asleep state, exercise, and cuff calibration stability. These recommendations provide valuable insights for future clinical studies and investigations in BP monitoring. They will play a major role in addressing current challenges, as highlighted in recent work[40], including individual cuff calibration, post-calibration measurement stability, tracking BP changes, and implementing machine learning algorithms in cuffless BP estimation. Further validation and refinement of these recommendations through clinical field investigations are essential for their real-world application. Additionally, investigating the practical implications and potential issues associated with the clinical implementation of cuffless BP readings is crucial to ensure their reliable and accurate use in clinical practice.

## Conclusion

Despite the growing interest in PPG-based models for estimating BP, navigating the current research landscape can be complex and confusing. With the lack of clear guidelines for model evaluation, and use of small datasets that may not represent the general population, there is a significant risk of measurement bias in PPG-based BP models for medical applications. Inconsistent use of performance measures and failure to apply ISO 81060-2:2018 further compound these issues, highlighting the need for new standards for non-invasive BP measurement. Current guidelines also fail to discuss the collection of training and testing data from pregnant women, further highlighting this need. To address these challenges and ensure consistent BP assessment for all, we make key recommendations for developing and deploying PPG-based BP models. By adhering to these guidelines, we can harness the full potential of PPG technology for blood pressure measurement across diverse populations. This approach not only promises to enhance clinical applications but also extends the utility of PPG for at-home monitoring, making it an invaluable tool for patients and healthcare providers alike.

## Data availability

All data supporting the findings of this study are available within the paper and its Supplementary Information. The data corresponding to Figs. 1 and 2 can be found in Supplementary Data 1.

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

## Author contributions

M.E designed and led the study. M.E., F.H., R.R.F., A.A., J.A., H.S. and C.M. conceived the study. All authors approved final manuscript.

## Funding

## Competing interests

The authors declare no competing interests.
