## [Peer Review File · Communications Medicine]

Reviewers' comments:

Reviewer #1 (Remarks to the Author):

This review attempted to present pros and cons of metrics used for the evaluation of the accuracy of PPG-based algorithms for blood pressure measurement.

Comments

1. Abstract – The authors state: “We recommend the use of seven performance metrics”. But only six are listed.... If they imply that the Bland-Altman plot is the seventh, then this is not a metric, but a plot.
2. Photoplethysmography is a technology used to obtain arterial waveforms. Absolute BP values are then provided usually after the calibration of the cuffless device using traditional BP values (obtained by oscillometry or more rarely by auscultation). Calibration of these devices renders them “BP tracking” devices and not “BP measuring” devices. This has an enormous impact on the validation procedures, metrics and protocols that should/can be used. This core issue is not addressed by the authors.
3. Main text – “Camera-based systems...and estimate BP”: References need to be provided here.
4. Two standards are currently available for the validation of cuffless devices: one by IEEE with an amendment (IEEE 1708–2014 & 1708a-2019) and one by ISO (ISO 81060-3:2022). IEEE uses mean absolute error and ISO uses mean error and SD of mean error. These are not discussed in the manuscript. A new ESH statement with requirements for validation cuffless BP devices is expected to be published in June 2023
5. Table 1 title says “Metrics” but Bland-Altman is a plot
6. Pearson correlation is an inappropriate metric to compare the agreement between two clinical measurement methods (<https://pubmed.ncbi.nlm.nih.gov/2868172/>).
7. Regarding the AAMI/ESH/ISO Universal Standard, it is stated: “Clinically acceptable”. However, this is inappropriate mainly because BP changes are not included in the validation procedures (<https://pubmed.ncbi.nlm.nih.gov/35708294/>).

Reviewer #2 (Remarks to the Author):

The manuscript offers a refreshingly candid review of the state of the emerging field of wearable devices for measuring blood pressure with photoplethysmography (PPG). The authors provide insightful comments on the main statistical methods used to assess performance of PPG devices and very clearly inform the reader with summaries of statistical features that, for instance, wearable device designers should include in performance tests. Two other aspects of this paper are of significance: (a) comments on subject study size (and cross-section) and (b) the use of standards. In the former, the authors point out that the tendency to test healthy subjects in many studies leaves

out an appropriate number of hypertensive, normotensive and hypotensive subjects that are needed for a device to be considered clinically relevant. In the latter case, the minimal (or absent) use of standards in device design creates intrinsic variance in device and algorithm performance that compromise relevance to narrow use cases. The manuscript offers highly constructive suggestions that any device or algorithm developer should closely examine and undertake their work.

Reviewer #1 (Remarks to the Author):

This review attempted to present pros and cons of metrics used for the evaluation of the accuracy of PPG-based algorithms for blood pressure measurement.

Comments

1. Abstract – The authors state: "We recommend the use of seven performance metrics". But only six are listed.... If they imply that the Bland-Altman plot is the seventh, then this is not a metric, but a plot.

Author response: Thank you for bringing to our attention the inconsistency in the performance metrics count in the abstract. As you correctly identified, we considered the Bland–Altman plot as the seventh, which is indeed not a metric but a plot. We appreciate this insight and will correct this for further clarity.

Author action 1: We modified the abstract to read as follows.

In this paper, we put forward essential recommendations for implementing a standardized error evaluation in the development of blood pressure models based on PPG. These recommendations are informed by a comprehensive analysis of seven widely utilized performance evaluation measures. We urge the utilization of seven performance measures to evaluate the efficacy of photoplethysmography (PPG)-based technologies in estimating blood pressure. These approaches comprise six quantitative metrics, namely mean absolute error, mean error (ME), standard deviation of the error (SDE), root mean squared error, Pearson's correlation coefficient, and absolute error of the sample. Additionally, we propose incorporating the Bland-Altman plot, a visual tool, to complement the quantitative metrics and facilitate a comprehensive assessment of the technology's performance.

Author action 2: We made the necessary substitution of the term "metric" with "measure" when appropriate and placed particular emphasis on distinguishing between quantitative measures and visual plots.

2. Photoplethysmography is a technology used to obtain arterial waveforms. Absolute BP values are then provided usually after the calibration of the cuffless device using traditional BP values (obtained by oscillometry or more rarely by auscultation). Calibration of these devices renders them "BP tracking" devices and not "BP measuring" devices. This has an enormous impact on the validation procedures, metrics and protocols that should/can be used. This core issue is not addressed by the authors.

Author response: Thank you for your insightful comment regarding the intrinsic functionality of PPG devices. Agreed.

Author action: To address this point, we added a new section "PPG-based device BP calibration"

\section*{PPG-based device BP calibration}

\subsection*{Importance of calibration in PPG-based BP estimation}

Calibration is a critical process that ensures accurate and reliable measurements in the context of non-invasive technology.\cite{jedrzejewski2021pulse,elgendi2019use} Calibration establishes a relationship between the PPG signal and reference BP measurements, aiming to obtain precise and absolute BP values. The reference BP measurements are typically obtained through direct invasive measurement using arterial BP lines (a cannula inserted into a peripheral artery) and alternatively the cuff-based sphygmomanometry.

In the development of a calibration model, the acquired PPG signals and their corresponding reference BP measurements are utilized. Various algorithms, such as regression models, machine learning techniques, and signal processing methods, are employed to establish a mathematical relationship or mapping between the features extracted from the PPG waveform and the BP readings. It is noteworthy that a significant portion of the research in this domain, aimed at creating algorithms that map PPG features to BP values,\cite{haddad2021continuous} or stratify BP using PPG features \cite{hu2023blood}.

\subsection*{Impact of anatomical site of PPG-based BP estimation}

The body location of PPG signal measurement plays a crucial role in calibration and BP estimation.\cite{elgendi2019use,butlin2018cuffless} Different body sites, such as the finger, wrist, or earlobe, can impact the morphology and characteristics of the PPG waveform. The distance between the measurement site and the heart introduces variations in the pulse transit time (PTT), which is the time for the arterial pulse wave to travel from the heart to the measurement site. The developed algorithms (i.e., calibration models) for BP estimation from the PPG often incorporate PTT as a feature or input.\cite{elgendi2019use} PTT serves as a proxy for BP estimation due to its correlations with BP. However, factors like vessel compliance, vascular tone, and anatomical distance between the measurement site and the heart can influence this relationship. To account for the impact of location on PPG-based BP estimation, calibration models may incorporate additional features or correction factors. These adjustments aim to normalize or account for the effects of the measurement site and its distance from the heart.

\subsection*{Calibration strategies for continuous PPG-based BP monitoring}

Calibration strategies for continuous PPG-based BP monitoring can be classified into two main approaches: subject-specific and non-subject-specific.\cite{aguet2023blood} Subject-specific calibration involves utilizing paired recordings of BP and PPG from an individual, providing a training phase for the algorithm with PPG data, enabling the generation of accurate BP estimations for subsequent continuous monitoring. Although subject-specific calibration offers high accuracy, it is not widely practical or cost-effective, with a device like Finapres (Finapres Medical Systems, Enschede, The Netherlands) being a prime example. On the other hand, non-subject-specific calibration incorporates data from multiple subjects, sometimes including pulse PTT measurements. However, it may have limitations in terms of accuracy. Therefore, many studies proposing PPG-based BP estimation opt for the non-subject-specific hybrid calibration strategy,

which combines information from diverse individuals to improve the accuracy and generalizability of the calibration model.\cite{aguet2023blood}

\subsection*{PPG-based BP tracking: Monitoring fluctuations over a period of time}

PPG-based BP tracking involves continuous monitoring or assessment of blood pressure over a period of time using photoplethysmography. This method observes changes in blood pressure trends relative to a baseline or previous readings and can be implemented through wearable fitness devices or smartphone apps that continuously measure and record PPG signals.

In a study conducted by Radha et al.\cite{radha2019estimating}, promising results were achieved in providing nocturnal systolic blood pressure tracking using a PPG-based wrist-worn device, with measurements taken every 5 seconds to produce an average BP value. However, their research did not include a discussion on calibration. Another approach to PPG-based BP tracking involves stratifying the data over different time intervals, such as 2 seconds\cite{liang2018hypertension} or 5 seconds \cite{hu2023blood, liang2018photoplethysmography}, and labeling the readings as hypertension, prehypertension, or normotension.

The information gathered from PPG-based BP tracking can be valuable for identifying patterns, trends, or fluctuations in blood pressure throughout the day or during specific activities, helping individuals become more aware of their blood pressure patterns and potential triggers for changes. However, it's important to note that while BP tracking through PPG offers valuable insights over time, it may not provide the most precise and absolute measurement of blood pressure at a given moment. For medical diagnosis and treatment decisions, the use of well-calibrated devices that provide absolute BP measurements is crucial.

\subsection*{PPG-based BP measuring: Precise point-in-time measurements}

PPG-based BP measuring involves obtaining precise and absolute blood pressure values at a specific point in time using photoplethysmography. This is typically done using validated and calibrated blood pressure measurement devices, such as mercury sphygmomanometers or arterial blood pressure (ABP) monitors, which provide readings in millimeters of mercury (mmHg) and offer accurate measurements of both systolic and diastolic blood pressure.

PPG-based BP measuring employs two main approaches: subject-specific and non-subject-specific. In a study by Tang et al.\cite{tang2022subject}, they attempted to map the PPG to ABP as a continuous signal using subject-specific calibration. The first 80\% of the PPG recording was utilized for training, while the remaining 20\% was used for testing. Similarly, the non-subject-specific approach to map PPG to ABP, as demonstrated by Ibtehaz et al.\cite{ibtehaz2022ppg2abp}. However, both approaches still face challenges in achieving real-time applications.

PPG-based BP measuring plays a crucial role in clinical settings, enabling healthcare professionals to diagnose and manage conditions like hypertension (high blood pressure) and hypotension (low blood pressure). These absolute measurements empower doctors to make informed decisions about treatment plans and medication adjustments.

3. Main text – "Camera-based systems...and estimate BP": References need to be provided here.

Author response: Thank you for your valuable comment. We agree with the importance of providing proper references when discussing the principles of camera-based systems used to detect blood pressure.

Author's action: We added two references to this sentence.

"Camera-based systems\cite{Ref1} use a camera to capture the image of the face or the finger and then analyze the pulsatile changes in skin color to estimate blood flow and BP. Radar-based systems\cite{Ref2} use radio waves to detect changes in blood volume and estimate BP."

Ref1. Djeldjli, Djamaledine, et al. "Remote estimation of pulse wave features related to arterial stiffness and blood pressure using a camera." *Biomedical Signal Processing and Control* 64 (2021): 102242.

Ref2. Buxi, Dilpreet, Jean-Michel Redouté, and Mehmet Rasit Yuce. "Blood pressure estimation using pulse transit time from bioimpedance and continuous wave radar." *IEEE Transactions on Biomedical Engineering* 64.4 (2016): 917-927.

4. Two standards are currently available for the validation of cuffless devices: one by IEEE with an amendment (IEEE 1708–2014 & 1708a-2019) and one by ISO (ISO 81060-3:2022). IEEE uses mean absolute error and ISO uses mean error and SD of mean error. These are not discussed in the manuscript. A new ESH statement with requirements for validation cuffless BP devices is expected to be published in June 2023

Author response: We thank the reviewer for thoughtfulness and valuable feedback. We did not though include the ISO 81060-3:2022 and ISO 81060-2 for the following reasons.

- The ISO 81060-3:2022 standard ("Non-invasive sphygmomanometers - Part 3: Clinical investigation of continuous automated non-invasive sphygmomanometers") does not provide guidelines or standards for conducting clinical investigations or assessments related to the mentioned types of sphygmomanometers or invasive blood pressure monitoring equipment.
- ISO 81060-2 is already using and citing ISO 81060-2:2018.

Author action 1: We added the following sentence at the end of discussion.

Our recommendations, in conjunction with the 2023 ESH statement\cite{stergiou2023european}, provide a comprehensive framework for validating cuffless blood pressure measuring devices. The ESH statement introduces six validation tests that assess key aspects including absolute blood pressure accuracy, robustness against hydrostatic pressure effects, accuracy during treatment, awake/asleep state, exercise, and cuff calibration stability.

Author action 2: We changed the title.

"Recommendations for Evaluating Photoplethysmography-based Blood Pressure Models"

5. Table 1 title says "Metrics" but Bland-Altman is a plot

Author response: Thank you for your insightful comment. The Bland-Altman plot is indeed not a numeric metric but a graphical method for comparing two measurement techniques. However, it has

been incorporated into Table 1 due to the ISO 81060-2:201811 requirement for its inclusion in clinical acceptance studies.

Author action: We addressed this point above.

6. Pearson correlation is an inappropriate metric to compare the agreement between two clinical measurement methods (<https://pubmed.ncbi.nlm.nih.gov/2868172/>).

Author response: We sincerely thank the reviewer for the valuable feedback.

Author action: We modified the text and added the suggested reference.

Pearson's correlation coefficient (R) measures the relationship between measured blood pressures from a reference device (e.g., arterial blood pressure) and the blood pressure estimated from the PPG-based BP values in continuous monitoring. It quantifies the strength and direction of the linear relationship, indicating the level of agreement between PPG-based blood pressure estimates and the reference measurements. Calculating the correlation coefficient allows for assessing the association between the two blood pressure measurements in the time series data. While R measures the linear relationship between two variables, it may not capture all aspects of agreement, especially if the relationship is nonlinear or there are systematic differences between the methods. In such cases, alternative metrics like Bland-Altman analysis may provide more comprehensive insights into the agreement between clinical measurement methods.

Bland, J. M. & Altman, D. Statistical methods for assessing agreement between two methods of clinical measurement. TheLancet 327, 307–310 (1986).

7. Regarding the AAMI/ESH/ISO Universal Standard, it is stated: "Clinically acceptable". However, this is inappropriate mainly because BP changes are not included in the validation procedures (<https://pubmed.ncbi.nlm.nih.gov/35708294/>).

Author response: We acknowledge the reference you provided.

Author action 1: We removed the term "clinically acceptable."

Author action 2: We added the following text clarify this point further.

These recommendations provide valuable insights for future clinical studies and investigations in blood pressure monitoring. They will play a major role in addressing current challenges, as highlighted in recent work (Stergiou2022cuffless), including individual cuff calibration, post-calibration measurement stability, tracking blood pressure changes, and implementing machine learning algorithms in cuffless blood pressure estimation. Further validation and refinement of these recommendations through clinical field investigations are essential for their real-world application. Additionally, investigating the practical implications and potential issues associated with the clinical implementation of cuffless blood pressure readings is crucial to ensure their

reliable and accurate use in clinical practice.

Stergiou, G. S. et al. Cuffless blood pressure measuring devices: review and statement by the European society of hypertension working group on blood pressure monitoring and cardiovascular variability. *J. hypertension* 40, 1449–1460 (2022).

Reviewer #2 (Remarks to the Author):

The manuscript offers a refreshingly candid review of the state of the emerging field of wearable devices for measuring blood pressure with photoplethysmography (PPG). The authors provide insightful comments on the main statistical methods used to assess performance of PPG devices and very clearly inform the reader with summaries of statistical features that, for instance, wearable device designers should include in performance tests. Two other aspects of this paper are of significance: (a) comments on subject study size (and cross-section) and (b) the use of standards. In the former, the authors point out that the tendency to test healthy subjects in many studies leaves out an appropriate number of hypertensive, normotensive and hypotensive subjects that are needed for a device to be considered clinically relevant. In the latter case, the minimal (or absent) use of standards in device design creates intrinsic variance in device and algorithm performance that compromise relevance to narrow use cases. The manuscript offers highly constructive suggestions that any device or algorithm developer should closely examine and undertake their work.

Author response: We greatly appreciate your positive feedback and careful evaluation of our manuscript.

REVIEWERS' COMMENTS:

Reviewer #1 (Remarks to the Author):

No further comments